

**The importance of cylinder passivation chemistry for preparation and long-term**
**stability of multicomponent monoterpene primary reference materials**
Nicholas D. C. Allen[1], David R. Worton[1], Paul J. Brewer[1], Celine Pascale[2], Bernhard
Niederhauser[2]
[1] National Physical Laboratory, Hampton Road, Teddington, TW11 0LW, UK.
[2] Federal Institute of Metrology METAS, Lindenweg 50, CH-3003 Bern, Switzerland.
Corresponding author:
nick.allen@npl.co.uk
Tel: +44 (0)20 8943 6913
**Abstract**
Monoterpenes play an important role in atmospheric chemistry due to their large anthropogenic
and biogenic emission sources and high chemical reactivity. As a consequence, measurements
are required to assess how changes in emissions of monoterpenes impact air quality. Accurate
and comparable measurements of monoterpenes in indoor and outdoor environments require
gaseous primary reference materials (PRMs) that are traceable to the international system of
units (SI). PRMs of monoterpenes are challenging to produce due to the high chemical
reactivity and low vapour pressures of monoterpenes and also their propensity to convert into
other compounds, including other terpenes.

In this paper, the long-term stability of gravimetrically prepared static monoterpene PRMs
produced in differently passivated cylinders, including sampling canisters, was assessed. We
demonstrate that static PRMs of multiple monoterpenes can be prepared and used as a suitable
long-term standard. For the first time the effect of cylinder pressure and decanting from one
cylinder to another on the chemical composition and amount fraction of monoterpenes was also
studied. Gravimetrically prepared PRMs of limonene in high pressure cylinders were compared
to a novel portable dynamic reference gas generator based on dilution of pure limonene vapour
emitted from a permeation tube.

**Keywords**
Monoterpenes, gravimetrically prepared standards, long-term stability, cylinder passivation,
sampling vessels, dynamic standard system, permeation, primary reference material

**1. Introduction**

Terpenes are a large and diverse family of naturally occurring organic compounds that are a
major biosynthetic building block (de Meijere, Hadjiarapoglou et al. 1998, Nicklaus, Minnaard
et al. 2013). Vegetation including forests and agricultural crops (Ormeño, Gentner et al. 2010,
Curtis, Helmig et al. 2014, Gentner, Ormeno et al. 2014) emit substantial quantities of isoprene
(a hemiterpene ($C_5H_8$)), monoterpenes ($C_{10}H_{16}$) and sesquiterpenes ($C_{15}H_{24}$) (Tao and Jain
2005, Barkley, Palmer et al. 2008, Smolander, He et al. 2014, Squire, Archibald et al. 2014,
Jokinen, Berndt et al. 2015).

Terpenes play an important role in atmospheric chemistry due to their high reactivity
influencing the $HO_x$ and $NO_x$ budgets (Presto, Hartz et al. 2005, Ng, Chhabra et al. 2007,
Forester and Wells 2011, Carslaw, Fletcher et al. 2017). The photochemical reactions of



terpenes can lead to the production of tropospheric ozone, which is highly toxic to humans
(Wolkoff, Clausen et al. 2000), and the formation of secondary organic aerosol with
implications for climate (Lee, Goldstein et al. 2006, Ng, Chhabra et al. 2007, Vibenholt,
Norgaard et al. 2009) that influence climate (Coleman, Lunden et al. 2008).
Terpenes are also known to be emitted from building materials (Allen, Brewer et al. 2016) and
household products, in which they are primarily used as fragrances and flavourings (Wolkoff,
Schneider et al. 1998, Lamorena and Lee 2008, Steinemann, MacGregor et al. 2011, Wang,
Barratt et al. 2017), impacting indoor air quality (Singer, Destaillats et al. 2006, Nazaroff and
Goldstein 2015). In particular, the exposure of the public to terpenes in indoor air quality is
poorly understood due to a lack of available data, despite the toxicity of their photochemical
products (Jones 1999, Wolkoff and Nielsen 2001, Wang, Ang et al. 2007, Wang, Barratt et al.
62  2017).

A variety of techniques have been used for the sampling and analysis of complex mixtures of
terpenes (Batterman, Zhang et al. 1998, Koch, Winterhalter et al. 2000, Birmili, Berresheim et
al. 2003, Pollmann, Ortega et al. 2005, Qiu, Smuts et al. 2017). However, the accurate
measurement of terpene amount fractions in indoor and outdoor air is highly dependent upon
the availability of appropriate SI traceable gaseous PRMs (Rhoderick 2010).
The World Meteorology Organisation (WMO) Global Atmosphere Watch (GAW) programme
is a framework to provide reliable scientific data and information on the long-term trends in
the chemical composition of the atmosphere. In WMO-GAW Report No. 171 Global Long-
Term Measurements of Volatile Organic Compounds (VOCs) new data quality objectives were
created for priority VOC compounds including monoterpenes. These data quality objectives
stipulated 20 % accuracy and 15 % precision for monoterpene measurements reported by GAW
stations. Further recommendations by GAW's scientific advisory group for reactive gases have
been made to lower these data quality objectives to 5 % accuracy (Hoerger, Claude et al. 2015).
In order to meet the 5% uncertainty target and prevent the reference material dominating the
uncertainty requires stable PRMs of monoterpenes with uncertainties of better than 1 %. There
is also a requirement for performing reliable sampling or dynamic calibration methods for the
in-situ calibration of instruments during field campaigns or at long-term atmospheric
monitoring stations and for independent verification of the gaseous PRMs.
PRMs containing monoterpenes are challenging because monoterpenes are highly reactive
compounds and can isomerise, tautomerise or react to form a wide range of other compounds
including other terpenes (Findik and Gunduz 1997, Allahverdiev, Gunduz et al. 1998, Foletto,
Valentini et al. 2002). This has led to observations that the amount fraction of some
monoterpenes increase overtime, including the observation of compounds that were not present
when the mixture was first prepared, while the amount fraction of others decline (Rhoderick
and Lin 2013). Moreover, cylinder passivation has a big impact on the stability of monoterpene
gas mixtures. Rhoderick and Lin (2013) demonstrated that specific passivation types, such as
'Experis' (Quantum) manufactured by Air Products, looked the most promising for
monoterpenes.
In this paper, multicomponent monoterpene static gaseous PRMs containing α-pinene, 3-
carene, R-limonene and 1,8-cineole, as well as a mixture containing β-pinene were prepared
gravimetrically at high pressure in cylinders with different surface passivations (the coating
applied to the internal surface of a cylinder to reduce adsorptive losses). The effects of
adsorption to the cylinder walls and the cylinder pressure were assessed through a series of



decanting experiments for these different cylinder passivations. The monoterpene PRM in the
most suitable cylinder passivation treatment was analysed repeatedly over a two year period to
assess the long-term stability building on the previous shorter-term stability studies of
Rhoderick et al. (Rhoderick 2010, Rhoderick and Lin 2013). The PRM containing limonene
was compared to a new dynamic system based on permeation known as Reactive Gas Standard
2 (ReGaS2) developed by the Federal Institute of Metrology (METAS), (Pascale, Guillevic et
al. 2017), that is based on permeation to evaluate any systematic biases between the two
different approaches. A portion of a monoterpene PRM was decanted into a SilcoNert 2000®
(Silcotek) treated sampling canisters to study the stability and their suitability for short-term
storage after field sampling.

**2.   Experimental**
**2.1. Gravimetric preparation of PRMs**
PRMs containing the four monoterpenes, $\alpha$-pinene (both the minus and plus optical isomers),
3-carene, *R*-limonene and 1,8-cineole and *n*-octane, that was used as an internal reference
standard, were prepared independently, in a balance of high purity dry nitrogen (BIP+, Air
Products) in accordance with ISO 6142 (ISO 2015). Each monoterpene compound was
prepared gravimetrically as a binary mixture (mixtures A – E) at an amount fraction of
nominally $5 – 10$ µmol mol$^{-1}$ by liquid injection of each monoterpene, via a transfer vessel, into
individual 10 L evacuated cylinders ($<4.0 \times 10^{-7}$ mbar). A balance of high purity dry nitrogen
(BIP+, Air Products) was added by direct filling through an additional purifier (Microtorr,
SP600F, SAES Getters) to remove trace impurities such as hydrocarbons and water to below
$< 1$ nmol mol$^{-1}$. Two $\beta$-pinene mixtures were also produced in a similar way (mixtures F and
G). The compound and the amount fraction of the parent PRMs were: limonene $4.968 \pm 0.044$
µmol mol$^{-1}$ (mixture A), $\pm$-$\alpha$-pinene $9.942 \pm 0.029$ µmol mol$^{-1}$ (mixture B), 1,8-cineole $5.007$
$\pm 0.028$ µmol mol$^{-1}$ (mixture C), 3-carene $4.954 \pm 0.036$ µmol mol$^{-1}$ (mixture D), octane $9.995$
$\pm 0.038$ µmol mol$^{-1}$ (mixture E), $\pm$-$\beta$-pinene $9.829 \pm 0.090$ µmol mol$^{-1}$ (mixture F) and $10.492$
$\pm 0.175$ µmol mol$^{-1}$ (mixture G) with all uncertainties in the gravimetric preparation expanded
($k = 2$).

All 'pure' liquid compounds were purchased from commercial suppliers (Fluka and Sigma
Aldrich) and were purity analysed following the guidelines stipulated in ISO 19229:2015 by
gas chromatography with a flame ionisation detector prior to use. Impurities were identified
and quantified by percentage area. The purity of all the monoterpenes was between 93.5 and
99.5 % (Table S1, supporting information).

A PRM of nominally 100 nmol mol$^{-1}$ (mixture AA, see Table 1) containing the four
monoterpenes and *n*-octane was prepared by direct transfer of a portion (10 – 25 g) of each
gravimetrically prepared parent mixture (A – E) and topped up with a balance of filtered high
purity dry nitrogen (BIP+, Air Products) that was again added by direct filling through the
Microtorr, SP600F, SAES Getters. A final dilution stage was carried out to prepare a PRM at
nominally 2 nmol mol$^{-1}$ (mixture BB, Table 1). A second nominal 2 nmol mol$^{-1}$ mixture
(mixture CC) was prepared in the same way to mixture BB for the long-term stability
comparison. All of the PRMs were prepared in 10 litre Experis passivated cylinders from Air
Products, Belgium.





**Table 1.** Gravimetric compositions of monoterpene PRMs made by dilution of the parent mixtures (mixtures A-E). Amount fractions are all in nmol mol$^{-1}$, uncertainties in the gravimetric preparation are expanded ($k = 2$) and do not include uncertainties arising from the verification process.

| Compound | Cylinder assignment | | |
|---|---|---|---|
| | **AA** | **BB** | **CC** |
| **limonene** | $93.1 \pm 0.8$ | $2.009 \pm 0.018$ | $2.042 \pm 0.019$ |
| *α*-**pinene** | $96.1 \pm 0.8$ | $2.075 \pm 0.019$ | $2.109 \pm 0.019$ |
| **1,8-cineole** | $94.2 \pm 0.5$ | $2.033 \pm 0.012$ | $2.066 \pm 0.012$ |
| **3-carene** | $91.1 \pm 0.7$ | $1.967 \pm 0.015$ | $1.999 \pm 0.015$ |
| *n*-**octane** | $89.0 \pm 0.46$ | $1.920 \pm 0.010$ | $1.952 \pm 0.010$ |

## 2.2. Analytical set-up

All of the measurements were performed using a gas chromatograph (Varian CP-3800) with an FID. The system uses a sample pre-concentration trap containing glass beads cooled by liquid nitrogen and held at -100 °C during sampling to collect and focus the analytes prior to injection and separation on a GC column (Varian CP-Sil 13; 75 m x 0.53 mm, phase thickness = 2.0 µm). All mixtures were connected to the GC using SilcoNert 2000® passivated 1/16″ stainless steel tubing. The lines were thoroughly purged and flow rates were allowed to stabilise for at least 10 minutes before commencing analysis.

The PRMs were connected using a minimal dead volume connector and the flow rate was set to 50 ml min$^{-1}$ using a custom flow restrictor. For the dynamic ReGaS2 system a flow of 50 ml min$^{-1}$ could not be achieved and the volume flowed across the trap was recorded by a mass flow meter, calibrated with BIP+ nitrogen, and subsequently corrected to match the sample volume of the high pressure gas standards. Mixtures were compared by running a series of up to six replicate analyses in blocks with the unknown mixture being analysed between two blocks of the PRM mixture to correct for any instrumental drift during analysis. The observed relative standard deviations in the peak areas of all compounds were between 0.3 – 1.5 %.

## 2.3. Decanting experiments

A schematic illustrating the decanting procedure is shown in Figure 1. The decanting experiments were performed in 10 L aluminium Luxfer cylinders that had been treated with different types of cylinder passivation, these included Experis, sometimes referred to as as Quantum, (Air Products), SPECTRA-SEAL (BOC) and 'in-house' treated BOC SPECTRA-SEAL. It has been observed that this propriety 'in-house' passivation provides improved stability for a wide range of compounds at low amount fractions. All cylinders had a 10 L internal volume. Initially, a new PRM, identified as cylinder 1 in Figure 1 was prepared gravimetrically (as described in Section 2.2) at an amount fraction of nominally 2 nmol mol$^{-1}$ and a pressure of 120 bar (cylinder '1') from a dilution of a 100 nmol mol$^{-1}$ PRM (mixture AA).

Once a new PRM (cylinder '1') had been prepared at 120 bar (day 1), the mixture was analysed by GC-FID and compared against the reference PRM, mixture BB (day 2). The following day (day 3) approximately 50 bar of cylinder '1' was decanted by direct fill (a short well-purged transfer line) to cylinder '2' leaving 70 bar in cylinder '1'. Both cylinder '1' and '2' were then



analysed by GC-FID and compared against reference PRM, mixture BB. Finally (day 4),
approximately 20 bar of cylinder '2' was decanted to cylinder '3' leaving 30 bar in cylinder '2'
and both cylinder '2' and '3' were then analysed by GC-FID and compared against reference
PRM, mixture BB (differences in the gravimetric values between the PRM and the reference
standard were normalised). All of the cylinders were evacuated and the decant procedure was
repeated for a second time.
All of the analyses was performed using GC-FID as described in Section 2.2. The certified
valve of the decanted cylinder was determined through a comparison with a nominal 2 nmol
$mol^{-1}$ reference PRM (mixture BB). If there were no losses then the certified amount fraction
of the decanted cylinders should be the same as those of the PRM cylinder '1'. Decant losses
were determined for each compound by calculating the response factor ($RF_{decant}$) for the
decanted mixture:
$$RF_{decant} = \frac{Area_{avgdecant}}{Grav_{decant}}$$

where, $Area_{avdecant}$ is the average peak area for a set of GC runs of the decanted mixture and
$Grav_{decant}$ is the gravimetrically assigned value of the compound. The response factor of the in-
house reference PRM, mixture BB was also determined by:
$$RF_{BB} = \frac{Area_{avgBB}}{Grav_{BB}}$$

where, $Area_{avgBB}$ is the average peak area for a set of GC runs of in-house reference PRM,
mixture BB and $Grav_{BB}$ is the gravimetrically assigned value of the compound in mixture BB.
To determine the effects of decanting, results were normalised to take into account the
gravimetric difference between the in-house reference PRM (mixture BB) and the decanted gas
mixture and the difference between the areas were determined by:
$$Normalised\ ratio = \frac{RF_{decant}}{RF_{BB}}$$

From the normalised ratio percentage differences between the in-house reference PRM
(mixture BB) and the decanted mixture were determined.



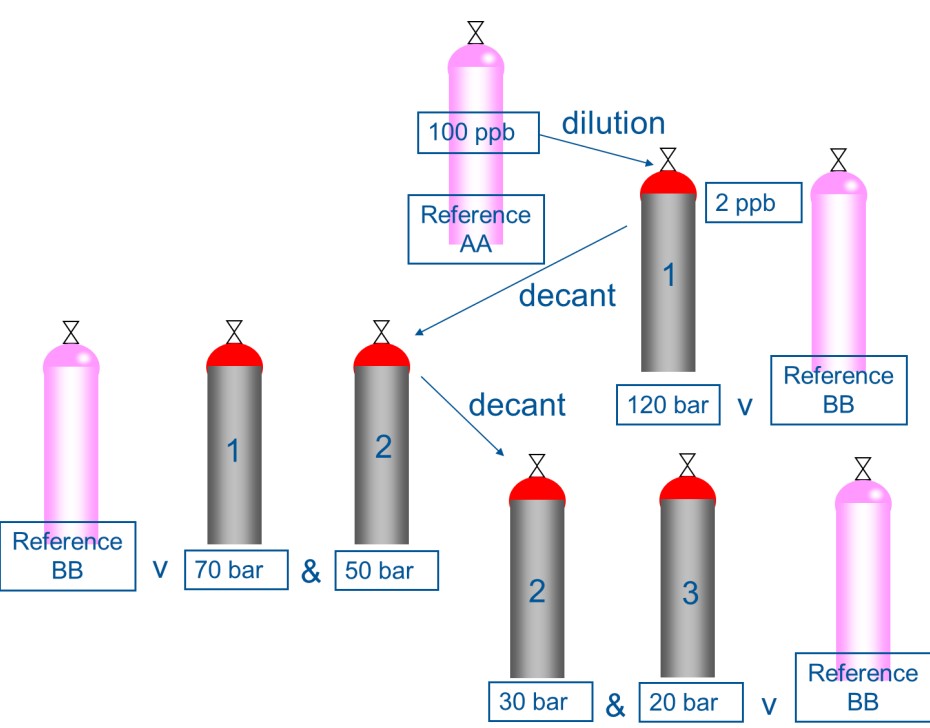

**Figure 1**: Schematic of the decanting procedure that was performed for the monoterpenes using 10 L Luxfer cylinders treated with different passivation types (Experis, SPECTRA-SEAL and an in-house treated SPECTRA-SEAL).

## 2.4. Short and long-term stability study of monoterpene PRMs

To determine the short and long-term stability of the four component monoterpene reference PRM, mixture BB was regularly run over a three month (75 day) period. GC peak area responses of each terpene were ratioed to *n*-octane, which is known to be stable in this passivation type for more than two years (Grenfell, Milton et al. 2010) and was present in the mixtures as an internal standard. The long-term stability of mixture BB (prepared on 2nd June 2015) was determined by preparing a fresh nominal 2 nmol mol$^{-1}$ mixture (mixture CC), prepared two and a half years later (904 days) on the 22nd November 2017, and comparing the peak areas and their response factor.

*β*-Pinene, which is known to decompose in the presence of other terpenes over time (Foletto, Valentini et al. 2002), was prepared at 10 µmol/mol in 2015. An independently prepared *β*-pinene binary was prepared two and a half years later and the areas and response factors were compared to determine stability.

## 2.5. Canister experiment

A large number of samples are collected in the field during measurement campaigns. It is imperative that these samples can be collected and stored in a way that preserves the contents





until they are analysed. One commonly used option is the use of sampling canisters or vessels
that have been evacuated prior to use. It has been well documented that the use of stainless
steel canisters for sampling terpenes can be problematic (Batterman, Zhang et al. 1998,
Rhoderick 2010). Here we decant a portion of our 2 nmol mol$^{-1}$ in-house reference PRM
(mixture BB) into a SilcoNert 2000® treated 2 L sampling canister to determine their suitability
for short-term storage of monoterpenes. The content was analysed by GC-FID and compared
against the same nominal 2 nmol mol$^{-1}$ reference PRM (mixture BB) to determine if any losses
were observed over a three month period (83 days).

### 2.6. ReGaS2 dynamic system

An alternative to PRM preparation in high pressure cylinders is dynamic preparation using
permeation. The ReGaS2 is a mobile generator that can produce traceable reference gas
mixtures, including terpenes (Pascale, Guillevic et al. 2017).
The method is based on permeation and subsequent dynamic dilution: a permeation tube
containing the pure terpene is stored in an oven used as permeation chamber. The pure
substance permeates at a constant rate into the matrix gas and was diluted to give the desired
amount fraction. The mass loss over time of the permeation tube is precisely calibrated using
a traceable magnetic suspension balance. All parts in contact with the reference gas were coated
with SilcoNert2000®.
The ReGaS2 mobile gas generator was fitted with a limonene permeation tube and set to
dynamically generate an output of nominally 4 nmol mol$^{-1}$. The amount fraction of the
limonene produced by the dynamic system was measured using the same analytical set-up as
was described in Section 2.2 and compared to our nominal 2 nmol mol$^{-1}$ reference PRM
(mixture BB).

### 2.7. Uncertainty calculations

The evaluation of measurement uncertainty was in accordance to the 'Guide to the expression
of uncertainty in measurement' (Joint Committee for Guides in Metrology 2008).
Below is a description of an uncertainty evaluation when comparing the response of an
unknown mixture against a validated calibration standard e.g. a PRM:

$$\bar{r} = \frac{2A_{u,avg}}{(A_{s,avg1} + A_{s,avg2})}$$

Where $\bar{r}$ is the average ratio, $A_{u,avg}$ is the average peak area from n repeated measurements of
the comparison mixture, $A_{s,avg1}$ is the average peak area from n repeated measurements of the
calibration standard before running the comparison mixture and $A_{s,avg2}$ is the average peak area
from n repeated measurements of the calibration standard after running the comparison
mixture.




The amount fraction of the target component in the comparison mixture, $x_u$, is then calculated by:

$$x_u = x_s \bar{r}$$


Where $x_s$ is the amount fraction of the target component in the standard. The standard uncertainty of the measurand, $u(x_u)$, is calculated by:


$$\frac{u(x_u)}{x_u} = \sqrt{\frac{u(x_s)^2}{x_s^2} + \frac{u(\bar{r})^2}{\bar{r}^2}}$$


$u(x_s)$ is the uncertainty of the reference standard $u(\bar{r})$ is the uncertainty of the ratio, it includes e.g. repeatability, internal blanks, peak shape (error of integration), error in the sample volume. The uncertainty in $\bar{r}$ is calculated by:


$$\frac{u(\bar{r})^2}{\bar{r}^2} = \frac{u(A_{u,avg})^2}{A_{u,avg}^2} + \frac{u(A_{s,avg1})^2}{(A_{s,avg1} + A_{s,avg2})^2} + \frac{u(A_{s,avg2})^2}{(A_{s,avg1} + A_{s,avg2})^2}$$


## 3. Results and discussion

### 3.1. Decanting experiments and selection of passivation treatment for long-term stability measurements

The adsorption of the monoterpenes to the internal surfaces of the cylinder and valve were investigated through a series of decanting experiments as detailed in Section 2.3. The results for the different passivation types at 120 bar are shown in Figure 2. There is a tabulated summary of the results of the decanting experiments in Tables S2 – S7.





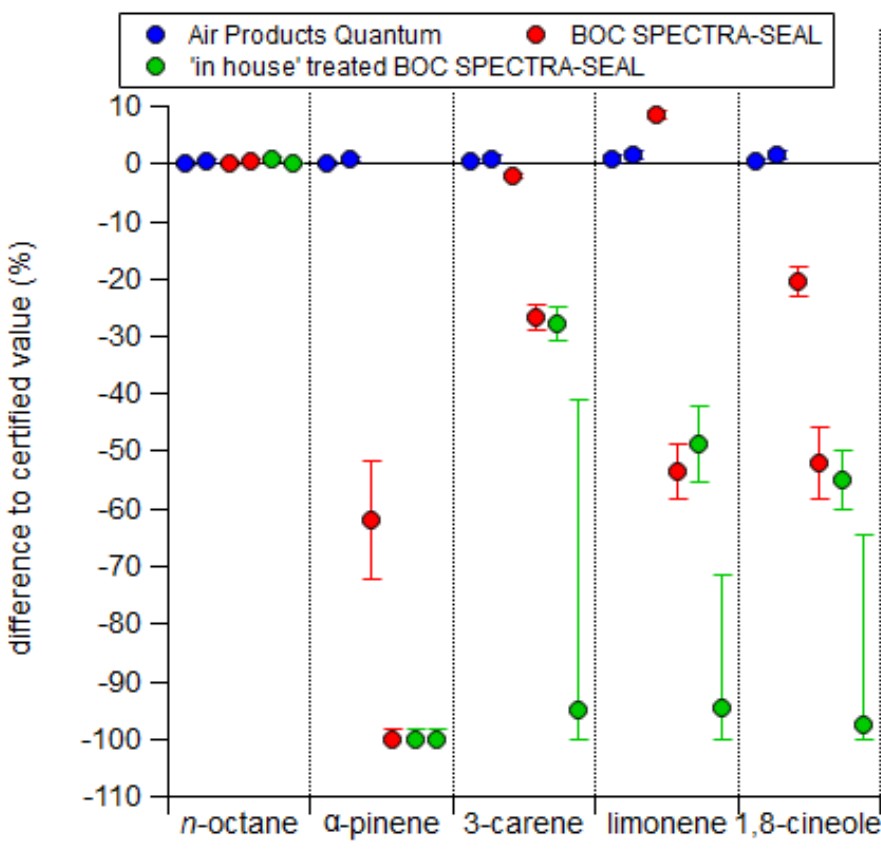

**Figure 2:** The relative difference between the certified amount fraction of the decanted
mixtures and the gravimetric amount fraction of the reference PRM (mixture BB). Each decant
was performed twice for each passivation type.
Decant losses of monoterpenes in the 10 L cylinders internally passivated with Air Products
Experis treatment were minimal (Tables S2 and S3). No obvious trends or patterns were
observed, therefore it can be confirmed, in agreement with Rhoderick et al. (Rhoderick and Lin
2013) that Experis cylinders are highly suitable for containing monoterpene PRMs. Figure 3
shows that the amount fraction does not appear to be strongly influenced by the pressure within
the cylinder, down to low pressure at 30 bar, as all agree within the measurement uncertainty
and there is no overall directional trend. Below 30 bar we observe that the ratio is less than 1
for all components. While the results are within the measurement uncertainty, wall factors
could have an influence on composition at low pressures (< 30 bar) (Brewer, Brown et al.
327 2018).
Figure 2 and Tables S3 and S4, show the initial decant, and repeat decant at 120 bar, in 10 L
cylinders passivated internally with BOC SPECTRA-SEAL treatment. Aside from the *n*-octane
a significant decrease in the amount fraction of all monoterpenes was observed (except for



limonene in the first decant) relative to the reference PRM (BB). No further decants were
performed for this cylinder type as the passivation was shown to be unsuitable for
monoterpenes, with strong degradation observed by gas chromatograph (Figure 4) within less
than 24 hours after making the initial PRM. In an attempt to improve the stability of trace
monoterpenes in SPECTRA-SEAL passivated cylinders, further in-house treatment was
applied to a new set of cylinders. The results of this are shown in Figure 2 and Tables S5 and
S6, however no improvement was observed and all of the monoterpenes showed significant
losses when the PRM was analysed by gas chromatograph, less than 24 hours after preparation.

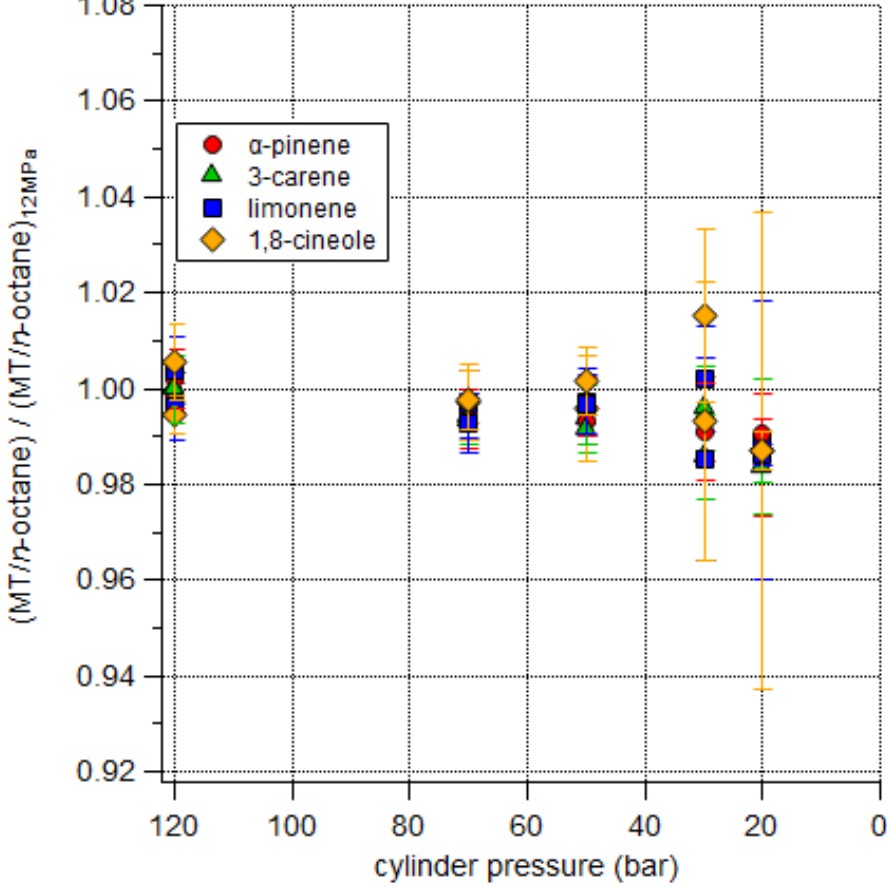

**Figure 3:** The relationship between cylinder pressure and monoterpene amount fraction after
normalisation to *n*-octane.
A sample of a monoterpene mixture in an internally treated SPECTRA-SEAL cylinder was
loaded onto a set of Chromasorb-106 and a set of Tenax sorbent tubes (both packed in-house)
and their contents analysed on a Thermal-Desorption Gas Chromatograph Mass Spectrometer
(TD-GC-MS) to identify the major degradation components. Similarly, a portion of the
reference PRM (mixture BB) was also loaded onto Chromasorb-106 and Tenax sorbent tubes
and analysed by TD-GC-MS. Five major peaks were consistently observed in the



chromatograms of the desorbed tubes (Figure 4). The peaks observed in the sample from the
SPECTRA-SEAL cylinder were identified as the following monoterpenes; (a) α-terpinene, (b)
τ-terpinene, (c) terpinolene, (d) cymene and (e) camphene based on retention time and MS
library matching to the NIST database. Mass spectrometry was used for compound
identification and good forward match (FM) and reverse match (RM) values, predominantly
>900 and all above 860 were obtained (see Tables S8 and S9 for details of the elution times,
FM and RM values and Figure S1 for mass spectra).
Interestingly, α-terpinene is produced industrially by acid-catalysed rearrangement of α-pinene
and camphene by oxidation of α-pinene (Findik and Gunduz 1997), which had disappeared
from the SPECTRA-SEAL passivated cylinders after 24 hours (Figure 4). A sample of the
reference PRM (mixture BB) was also loaded onto Chromasorb-106 and Tenax sorbent tubes.
No other terpenes or peaks were observed except for the expected α-pinene, 3-carene, R-
limonene and 1,8-cineole and n-octane. Kovats' Retention Indices were used to confirm the
assignment of terpene compounds (Table S10 and Figure S2).

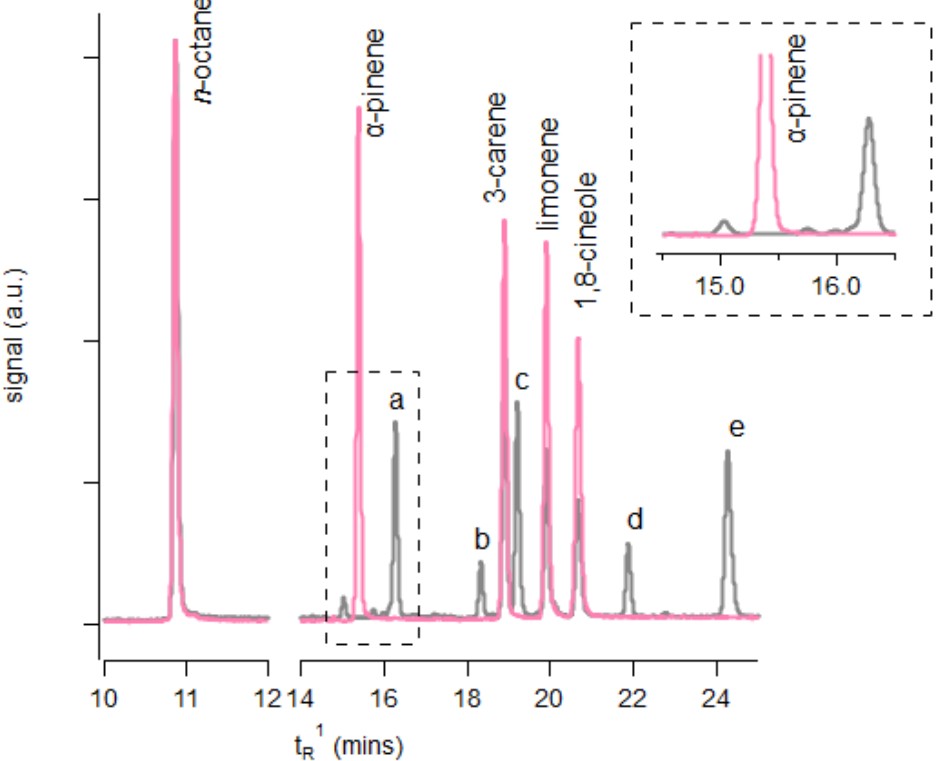

**Figure 4**: The nominally 2 nmol mol$^{-1}$ reference PRM, mixture BB (shown in pink) in an Air
Products Experis cylinder was made from the same parent PRM as the PRM made in the
internally passivated BOC SPECTRA-SEAL cylinder (shown in grey). The SPECTRA-SEAL
cylinder was analysed less than 24 hours after preparation and shows significant degradation
of the terpene compounds. The zoomed in portion of the chromatogram focuses on the α-pinene
peak (inset), showing that all of this compound has been lost. The additional peaks observed in





the analysis of the SPECTRA-SEAL passivated cylinder, labelled as a – e, correspond to those
named in the main text and to the observed MS shown in Figure S1.

**3.2. Short- and long-term stability study of monoterpene PRM**
The short-term and long-term stability of mixture BB was determined through a series of
experiments as detailed in Section 2.4. Over the first three month period that mixture BB was
run the ratio of the monoterpene to *n*-octane appears to remain fairly constant despite a few
outlying points.

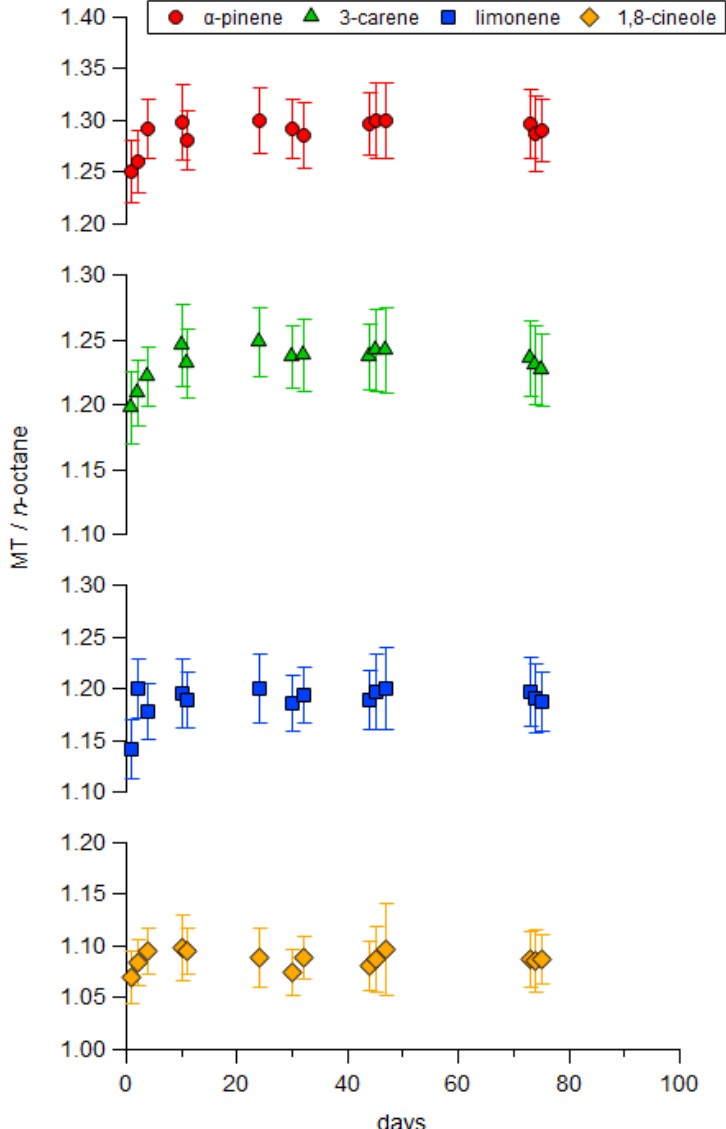






**Figure 5:** The short-term stability of reference PRM (mixture BB) at nominally 2 nmol mol$^{-1}$ compared as a ratio of the area of each monoterpene normalised relative to the $n$-octane internal standard. Error bars are included to account for the relative standard deviation of the mean ($k$ = 2).

Mixture BB was prepared on 2$^{nd}$ June 2015 and mixture CC was more than two years later (904 days) on the 22$^{nd}$ November 2017. A set of measurements were run to compare mixture BB and CC. This was repeated twice in the space of two days. Gravimetric values were normalised and the peak areas of the monoterpenes were then compared and the differences recorded (Table 2). It was found that, unsurprisingly, $n$-octane shows the best agreement and smallest difference, however all the monoterpenes agreed well and differences were no greater than 2.5 % between mixtures BB and CC. The relative standard deviations of the peak areas was between 0.1 – 1.5 % with the larger relative standard deviations correlating to the highest differences between the gas mixtures suggesting that the measurement is one of the largest sources of uncertainty in the experimental differences. The comparison infers that the monoterpene mixtures in Experis treated cylinders are stable for over two and a half years.

Mixture F and G containing $\beta$-pinene were prepared 976 days apart (approximately 2 years and 8 months difference), and were compared. Agreement for $\beta$-pinene, once normalised to take into account gravimetric differences, was better than 0.5 % and the relative standard deviation in the peak areas were 0.7 – 1.1 %. No systematic bias was observed. This suggests that in Experis treated cylinders there is little or no decay of $\beta$-pinene at the μmol/mol level when prepared as a binary mixture. Stability has been demonstrated for greater than two and a half years suggesting that it is the interaction of $\beta$-pinene with other monoterpenes in multicomponent gas standards that is the likely cause of their degradation.

**Table 2:** Comparison showing the percentage difference between PRM mixtures prepared more than two years apart to assess the long-term stability of mixture BB and mixture F. Gravimetric values were normalised and the peak areas compared.

| Compound | The difference when comparing PRMs | | |
|---|---|---|---|
| | Mixture BB v CC | | Mixture F v G |
| **limonene** | 0.24 % | 0.94 % | |
| **$\alpha$-pinene** | 0.06 % | 1.61 % | |
| **1,8-cineole** | 1.96 % | -0.22 % | |
| **3-carene** | -0.75 % | 1.35 % | |
| **$\beta$-pinene** | | | 0.45 % |
| **$n$-octane** | -0.75 % | 0.24 % | |

**3.3. Short-term stability of monoterpenes in treated sampling canisters**

Field campaign measurements require the short-term storage of VOC samples. Sampling canisters made from electropolished steel are frequently used despite losses being observed (Batterman, Zhang et al. 1998). Another solution is to use SilcoNert 2000® treated canisters (silanisation treatment, Silcotek). However, the SPECTRA-SEAL cylinders that performed poorly in the decant experiments, also use a silanisation surface treatment, therefore it was important to determine the suitability of SilcoNert 2000® treated canisters for short-term





storage of monoterpenes. Following decant of mixture BB into the SilcoNert 2000® treated
canister the contents were compared against mixture BB the following day (day 1) a week later
(day 8) and nearly three months later (day 83). The results of this are shown in Figure 6.

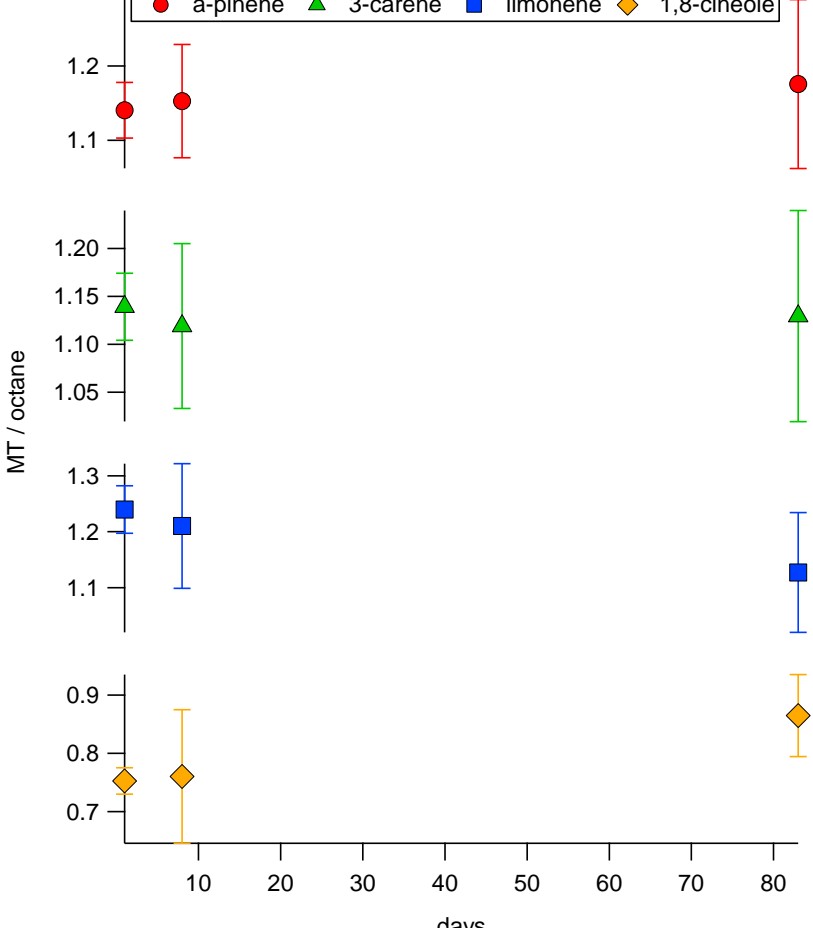

**Figure 6:** The short-term stability of mixture BB decanted into a SilcoNert 2000® treated
canister compared as a ratio of the area of each monoterpene normalised relative to the *n*-octane
internal standard. Error bars are included to account for the relative standard deviation of the
mean ($k = 2$).
No statistically significantly trends were observed for the stability although higher than normal
relative standard deviations in the GC peak areas were observed ($\leq 4$ % for all components
except 1,8-cineole which was $\leq 8\%$). This can be attributed to changes in the flow of gas from
the canister samples during measurement due to the volume and pressure of gas contained.
It appears that unlike the SPECTRA-SEAL passivated cylinders, the SilcoNert 2000® treated
canisters would allow the storage of multi-component monoterpene standards for up to three





months and still meet the data quality objective criteria recommended by GAW and its
scientific advisory group (Hoerger, Claude et al. 2015). Nevertheless, this does not mean that
a whole air sample containing terpenes or a broad array of terpenes together would behave in
the same way due to the impact of humidity, therefore more work is required to determine this.
However, it would suggest that decanting of PRMs for transport into the field in small
SilcoNert 2000® treated canister should be possible.

### 3.4. Comparison of dynamic and static PRM

From the weighing of the limonene permeation tube and from the data that was logged for the
nitrogen flow and subsequent dilution it was calculated that the ReGaS2 mobile gas generator
was outputting $4.41 \pm 0.32$ nmol mol$^{-1}$ of limonene with an expanded uncertainty of 7.3 % ($k$
$= 2$). Using the PRM static standards gravimetrically produced the output of the ReGaS2
dynamic system was estimated to be $3.57 \pm 0.11$ nmol mol$^{-1}$ of limonene with an expanded
uncertainty of 2.9 % ($k = 2$).

The static PRM that was used in this comparison (mixture BB) was also one of the mixtures
used as part of the CCQM-K121 monoterpene key comparison at nominally 2.5 nmol mol$^{-1}$
Results from CCQM-K121 demonstrated that all of the participants (Korea Research Institute
of Standards and Science, KRISS; National Institute of Standards and Technology, NIST and
NPL) agree within the $k = 2$ expanded uncertainties for all of the monoterpenes evaluated,
including limonene.

Reasons for the systematic bias between the two approaches are supposed, include the
temperature at which the permeator was operated was observed to have a strong influence on
the reproducibility of the permeation rate. At lower temperatures, such as 30°C (which was the
temperature used for the comparison), the permeator does not reach a true steady state and it
was observed that the variability on the permeation rate for the same temperature between two
measurements was between 8 and 10 %. A shift in the permeation rate of this magnitude
coupled to uncertainties in temperature would be enough to compensate for the systematic bias
observed between the two approaches.

The second reason is the decrease in the permeation rate: to investigate this further the
permeation rate of limonene from the ReGaS2 dynamic system was measured over an 11 month
period between March 2017 and February 2018. The decrease in the permeation rate was
determined to be 35 % over this temporal period (Figure S3) for the same temperature. The
measurement of the permeation rate in the magnetic suspension balance lasted between two
and seven days with an associated uncertainty between 0.5 and 1.5 % for one measurement at
one temperature.

A decrease in the permeation rate of this magnitude coupled to the high uncertainties at such
low temperatures would be enough to compensate for the systematic bias observed between
the two approaches. Despite the systematic bias observed between the two methods at this trace
level, the results of this first comparison are encouraging and show that state-of-the-art
developments are being made with dynamic systems capable of delivering reliable outputs
suitable for calibrating in the field systems.





## 4. Conclusions


In this paper we have investigated the short-term and long-term stability of monoterpenes in
differently internally passivated cylinders. The choice of cylinder passivation is critically
important in the preparation of monoterpene gas mixtures. We have demonstrated that Experis
treated cylinders are the most appropriate for containing low amount fraction monoterpene
PRMs and that the amount fraction is not influenced by pressure dependency between 120 and
30 bar.

The need for suitable storage and transport of PRMs into the field has driven us to investigate
the suitability of using SilcoNert 2000® treated canisters for monoterpenes. It was discovered
that SilcoNert 2000® treated canisters could hold monoterpenes for up to three months with
an uncertainty of 10 %, in line with GAW data quality objectives.

We compared the ReGaS2 dynamic mobile generator against high pressure static PRMs
gravimetrically prepared at NPL. It was found that the output of limonene from dynamic
ReGaS2 was 15 - 20 % lower than calculated. These differences correspond to less than 0.5
nmol mol$^{-1}$ and it has been suggested that the bias may be attributed to the reproducibility of
the limonene permeator at low temperature due to the permeation rate not reaching equilibrium.
This first comparison of a dynamic terpene standard against a traditional static standard is the
first step in providing the community with traceable reference materials suitable for in the field
measurements to meet GAW data quality objectives.

**Acknowledgements**
NPL and METAS were both funded as part of the European Metrology Research Programme
(EMRP) 'Metrology for VOC indicators in air pollution and climate change (KEYVOC)'.
The EMRP is jointly funded by the EMRP participating countries within EURAMET and the
European Union.

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
