# Peer review of "The importance of cylinder passivation for preparation and long-term stability of multicomponent monoterpene primary reference materials"

_Atmospheric Measurement Techniques, 2018_

## Referee Comment (RC1) · Anonymous Referee #1 · 9 Jul 2018

The manuscript entitled "The importance of passivation chemistry for preparation and long-term stability of multicomponent monoterpene primary reference materials" describes an investigation into the stability of monoterpenes in different gas cylinders and sample containers at nmol/mol levels.

The authors present evidence that several monoterpenes can be stored long-term in a particular type of cylinder. They also show that in other cylinders, with a different internal passivation treatment, the monoterpenes degraded considerably, and underwent oxidation or other coversion to different species. This is useful information and will help

inform researchers on choices of cylinder types for reference materials. The authors also show that reference materials containing monoterpenes can be sub-sampled, or transfered, to certain passivated stainless steel containers, which implies that these containers are useful for storing air samples or sub-sampling from high-pressure cylinders.

General Comments

The manuscript does not really address "chemistry" of passivation: I suggest removing "chemistry" from the title.

The citation format is annoying. Why are there two authors listed in many of the citations? This is not consistent with AMT style.

Section 2.7 could be moved to the supplement. The derivation of uncertainty is not critical to the manuscript.

The manuscript could use some minor editing for grammar. I've included some suggestions.

Specific Comments

P1, L40: Nicklaus et al. 2013 not found in reference list.

P2, L67: I would argue that accurate measurment is dependent on PRMs (Rhoderick 2010) AND analytical methods (Helmig et al. 2013).

P2, L79: Where does "better than 1%" come from? Is there a citation you can add? One might argue that a factor of 4 is required, the 5%/4 = 1.25%.

P2, L97: Move explaination of passiviation to first use of the term (previous paragraph).

P3, L114: Subject/verb agreement. Suggest splitting this sentence into two. "PRMs containing the four monoterpenes, $\alpha$-pinene (both the minus and plus optical isomers), 3-carene, R-limonene and 1,8-cineole were prepared independently in a balance of

high purity dry nitrogen (BIP+, Air Products) in accordance with ISO 6142 (ISO 2015). N-Octane was also added and used as an internal standard."

P 3, L 122: Were both hydrocarbons and water removed to < 1 nmol/mol, or just water?

P4, Table 1: Please explain what the "verification process" is.

P4 4, L 164: Again, consider two sentences. "... a flow of 50 mL/min could not be achieved. Consequently, the flowrate across the trap was recorded using a mass flowmeter, .... "

P5, L195: "value" or "amount fraction" instead of "valve". And is it really "certified"?

P7, L246: Please clarify if this applies to dry samples or natural air samples, for which some humidity can improve storage.

P7, L272: Suggest section 2.7 be moved to Supplement

P10, L346: This sentence should be re-arranged: "To investigate potential degradation components, a sample of a monoterpene mixture in an inernally treated SPECTRA-SEAL cylinder was loaded onto a sets of Chromasorb-106 and Tenax sorbent tubes (both packed in-house) and analysed on a Thermal-Desorption Gas Chromatograph Mass Spectrometer (TD-GC-MS)."

P10, L351: "Five additional peaks were consistently observed ...."

P11. L352: "The additional peaks ...."

P12, L382: replace "run" with "analyzed"

P13, L413: In Table 2, it was confusion at first that the two columns represent two different days. Why not just show one column of data, representing the average and standard deviation over 2 days. Otherwise it seems to the reader that there is something siginificant about the differences between the two days.

P14, L427: suggest: "were compared after 1, 8, and 83 days."

P14, L440: This needs more explanation. What was it about the sample flow: too little flow, too little pressure to flush tubing?

P15, L452: An introductory sentence is needed here. This paragraph should begin with the objective: comparing gravimetric PRMs to an SI-traceable dilution method.

P15, L462: semi-colon after NIST

P15, L466: poor grammar here.

P16, L506: This percentage difference should be stated in the previous section.

Also, if there were additional complications with the dilution system, the uncertanty assigned is probably too low.
* * *

---

## Referee Comment (RC2) · Anonymous Referee #2 · 23 Sep 2018

The manuscript "The importance of cylinder passivation chemistry for preparation and long-term stability of multicomponent monoterpene primary reference materials" by Allen and co-workers focuses on quantifying the long-term stability of gaseous monoterpene standards in a variety of commercially available cylinders. The authors find that a type of cylinder is suitable for storing these components, while others exhibit degradation after as little as 24 hours. The stability of the same species in sampling cylinders routinely used in the field is also investigated.

The method described in the manuscript is rigorous and the results are very important

for the broader atmospheric chemistry community. Monoterpene measurements are crucial to answer a number of open questions in tropospheric chemistry (missing OH reactivity, secondary aerosol formation, etc.), therefore establishing a solid metrological basis for these measurements is a very important and necessary step. However parts of the manuscript are unclear and would benefit from a few edits. I would therefore recommend publication on AMT once the points below are addressed.

Main points:

Lines 210-212: I find this paragraph confusing and after reading it many times I do not think it actually describes what you are doing with the normalised ratio. The crux of it all is the term 'gravimetric difference'. What you really are looking at here is the difference between the amount fraction you would expect in the absence of losses post-decant (in which case it would be equal to the gravimetric amount fraction of cylinder 1 as calculated from the dilution data) and the certified amount fraction obtained from a comparison with mixture BB as you described in lines 194-196. Please re-phrase.

Line 382 and Figure 5: One of the major outcomes of this study is the stability of the monoterpenes studied in one type of cylinder. This is shown in Figure 5 but this finding is backed up by surprisingly little statistical analysis. "appears [. . .] fairly constant" is the only description accompanying the plots in Figure 5. I feel that the conclusions of this study would be stronger if backed up by a more solid analysis. I would like the authors to add a weighted least-squares fit to the plots in Figure 5 and comment on the magnitude of the gradients in the light of its associated uncertainties from the fit. I would be surprised if this analysis did not support the authors' conclusions, but it needs to be shown. Also crop the x-axis at 80 days.

Minor points:

Lines 64-68: As you went through the effort of referencing work on terpene sampling, I feel it would be worth mentioning what these techniques actually are. I appreciate that a full account on terpene detection is beyond the scope of this manuscript, but it would

be nice to at least mention the techniques used by the papers cited.

Line 97: this is not the first occurrence of the word 'passivation' in the text. Please move the explanation of its meaning to line 90.

Line 106: "based on permeation" sounds too vague. Consider changing to something along the lines of "based on the dynamic dilution of limonene from a permeation tube".

Lines 114-115: you need to stress the difference between the 4 PRMs containing monoterpenes and the one containing n-octane. I recommend this edit: "PRMs containing the four monoterpenes, $\alpha$-pinene (both the minus and plus optical isomers), 3-carene, R-limonene and 1,8-cineole, as well as one containing n-octane (used as an internal reference standard), were prepared independently (no comma!) in a balance of high purity. . .".

Line 116: "high purity dry nitrogen". I feel like the inclusion of "dry" here (as well as in lines 120 and 140) is redundant, especially as the water content is specified further down (line 123).

Line 141: No point in repeating the specs of the purifier here (they already appeared in lines 121-122). Just replace with "purifier".

Line 162: connected to what?

Line 180 and following: I do not understand the use of the quote marks for the cylinder number here and in the rest of the manuscript. 1, 2 and 3 are the effective labels of the mixtures, just like AA and BB. I would recommend removing these as they are confusing and redundant.

Line 202: how many runs were averaged in each set? Give a typical value in the text

Lines 324-327: this is very interesting and perhaps deserves a short explanation of why such effects arise at low pressures?

Lines 362-363: The sentence starting with "A sample of the reference. . ." is a repetition

of the sentence in lines 349-350.

Line 396: where does the figure "2.5%" come from? The largest difference in Table 2 is 1.96 %. Perhaps change to "no greater than 2%"?

Lines 466-473: revise the grammar of this entire paragraph.

Table 2: explain in the table caption why there are two columns for BB v CC.

Figures 2, 3, 4, 5: These all seem to be at a lower resolution than Figures 1 and 6. Would the authors provide higher resolution figures? The better resolution helps a lot, especially when zooming in on the electronic version of the paper.

Figure 4: the caption needs to actually introduce what the figure represents. Start off with "Typical chromatogram of…"

Tables S2-S7: in the captions, state what the percentage difference refers to. The response factors? The normalised ratios?

Tables S2 and S3: data in S3 is the repeat of S2, but the cylinder types are labelled differently. I understand that Experis and Quantum refer to the same thing, but it would make things easier to just stick to one name (and it would also be consistent with the following tables).

Tables S4-S7: in the captions, state why there are missing entries in these tables.

Typographical, grammar and language corrections:

The citation style is not in the AMT recommended format. Revise throughout.

Shouldn't all equations be numbered?

Line 43: avoid double parentheses when possible e.g., a hemiterpene of formula C5H8

Lines 78-79: This sentence sounds like a fragment of a larger sentence. I suspect it should be part of the previous sentence. Please rephrase. Also, change 'prevent the reference material dominating' to 'prevent the reference material from dominating'.

Line 163: replace 'ml' with 'mL' (occurs twice in this line)

Line 175: no need for the comma after Quantum.

Line 194: Replace 'was' with 'were'

Line 195: Replace 'valve' with 'value'

Line 204: add a comma after BB.

Lines 207-208: remove the comma after "where" and add a comma after BB.

Line 228: replace 'run' with 'analysed' or similar.

Line 234: response factors (plural)

Line 256: replace with "can produce traceable reference gas mixtures of a number of species, including terpenes"

Lines 258-263: the verb tenses are all mixed up (past, present). It makes following the text harder than it should be. Please pick one tense and stick to it.

Lines 267-268: remove "as was"

Lines 353: replace semicolon with colon

Line 488: change to "systems in the field".

Line 496: I am not a huge fan of the word 'dependency'. 'Dependence' is more commonly used when describing the relationship of a variable to another (temperature-dependence, pressure-dependence, and so on).

Tables S2-S7: In one of the headers, change "difference to" with "difference with respect to"

Figure S1: in the caption, change spectrum to spectra and remove the comma after terpinolene.

[Figure]

---

## Author Comment (AC1) · 11 Oct 2018

The authors would like to express their gratitude to the Editor for handling the manuscript and to both of the Reviewers for their thoroughness and help in improving the quality of this manuscript with knowledgeable and constructive comments. We also thank the reviewers for underlining the need for metrology and in-situ sampling of monoterpenes by supporting this manuscript. We appreciate their time and effort in contributing. We hereafter provide a point by point response to their comments.

**Reviewer 1 Comments**

**General Comments**

The manuscript does not really address "chemistry" of passivation: I suggest removing "chemistry" from the title.

The word "chemistry" has been removed.

The citation format is annoying. Why are there two authors listed in many of the citations? This is not consistent with AMT style.

Apologies, the library was formatted in the wrong style – this has been corrected to be consistent with AMT style.

Section 2.7 could be moved to the supplement. The derivation of uncertainty is not critical to the manuscript.

The derivation of the uncertainty has been moved to the supplement.

The manuscript could use some minor editing for grammar. I've included some suggestions.

Specific Comments

P1, L40: Nicklaus et al. 2013 not found in reference list. Well spotted! It has been added.

P2, L67: I would argue that accurate measurment is dependent on PRMs (Rhoderick 2010) AND analytical methods (Helmig et al. 2013).

Sentence now reads: "However, the accurate measurement of terpene amount fractions in indoor and outdoor air is highly dependent upon the availability of appropriate SI traceable gaseous PRMs (Rhoderick 2010) and analytical methods (Helmig et al., 2013)."

P2, L79: Where does "better than 1%" come from? Is there a citation you can add? One might argue that a factor of 4 is required, the 5%/4 = 1.25%.

You are correct. Sentence now reads "In order to meet the 5 % uncertainty target and prevent the reference material from dominating the uncertainty requires stable PRMs of monoterpenes with uncertainties of better than 1.25 % (less than a quarter of the uncertainty)" – rather than one fifth/20 % as stated previously.

P2, L97: Move explaination of passiviation to first use of the term (previous paragraph).

This was moved.

P3, L114: Subject/verb agreement. Suggest splitting this sentence into two. "PRMs containing the four monoterpenes, α-pinene (both the minus and plus optical isomers), 3-carene, R-limonene and

1,8-cineole were prepared independently in a balance of high purity dry nitrogen (BIP+, Air Products) in accordance with ISO 6142 (ISO 2015). N-Octane was also added and used as an internal standard."

Following both Reviewers comments the sentence now reads: "PRMs containing the four monoterpenes, α-pinene (both the minus and plus optical isomers), 3-carene, R-limonene and 1,8-cineole, as well as one containing n-octane (used as an internal reference standard), were prepared independently in a balance of high purity dry nitrogen 115 (BIP+, Air Products) in accordance with ISO 6142 (ISO, 2015)."

P 3, L 122: Were both hydrocarbons and water removed to < 1 nmol/mol, or just water?

Yes, both. Sentence now reads: "A balance of high purity nitrogen (BIP+, Air Products) was added by direct filling through an additional purifier (Microtorr, SP600F, SAES Getters) to remove trace impurities to below < 1 nmol mol$^{-1}$, such as hydrocarbons and water."

P4, Table 1: Please explain what the "verification process" is.

Table caption now reads: "Gravimetric compositions of monoterpene PRMs made by dilution of the parent mixtures (mixtures A-E). Amount fractions are all in nmol mol-1, uncertainties in the gravimetric preparation are expanded ($k$ = 2) and do not include uncertainties arising from the experimental validation."

P4 4, L 164: Again, consider two sentences. "... a flow of 50 mL/min could not be achieved. Consequently, the flowrate across the trap was recorded using a mass flowmeter, .... "

The sentence has been separated into two, as recommended.

P5, L195: "value" or "amount fraction" instead of "valve". And is it really "certified"?

The typo has been amended. The authors accept that the word "certified" is misleading and this paragraph plus ensuing equations have been modified to address the points raised by both Reviewers.

P7, L246: Please clarify if this applies to dry samples or natural air samples, for which some humidity can improve storage.

Sentence now reads: "It has been well documented that the use of stainless steel canisters for sampling terpenes in dry or humidified air can be problematic (Batterman et al., 1998; Rhoderick, 2010)."

P7, L272: Suggest section 2.7 be moved to Supplement

The derivation of the uncertainty has been moved to the supplement. A sentence remains to explain that "The evaluation of measurement uncertainty was in accordance to the 'Guide to the expression of uncertainty in measurement' (Joint Committee for Guides in Metrology, 2008)."

P10, L346: This sentence should be re-arranged: "To investigate potential degradation components, a sample of a monoterpene mixture in an inernally treated SPECTRASEAL cylinder was loaded onto a sets of Chromasorb-106 and Tenax sorbent tubes (both packed in-house) and analysed on a Thermal-Desorption Gas Chromatograph Mass Spectrometer (TD-GC-MS)."

The Reviewer's sentence has been adopted and now reads: "To investigate potential degradation components, a sample of a monoterpene mixture in an  internally treated SPECTRA-SEAL cylinder

was loaded onto a set of Chromasorb-106 and Tenax sorbent tubes (both packed in-house) and analysed on a Thermal-Desorption Gas Chromatograph Mass Spectrometer (TD-GC-MS).

P10, L351: "Five additional peaks were consistently observed ...."

We think the word "additional" is misleading. Sentence reads: "Five major peaks were consistently observed in the chromatograms of the desorbed tubes (Figure 4)."

P11. L352: "The additional peaks ...."

Changed, the sentence reads: "The additional peaks observed in the sample from the SPECTRA-SEAL cylinder were identified as the following monoterpenes: (a) $\alpha$-terpinene, (b) $\tau$-terpinene, (c) terpinolene, (d) cymene and (e) camphene based on retention time and MS library matching to the NIST database."

P12, L382: replace "run" with "analyzed"

Sentence now reads: "Over the first three month period that mixture BB was **analysed** the ratio of the monoterpene to $n$-octane remains constant within the measurement uncertainty."

P13, L413: In Table 2, it was confusion at first that the two columns represent two different days. Why not just show one column of data, representing the average and standard deviation over 2 days. Otherwise it seems to the reader that there is something siginificant about the differences between the two days.

If the data was averaged it would not account for drift. However, Table 2 caption has been clarified to "Comparison showing the percentage difference between PRM mixtures prepared 389 more than two years apart to assess the long-term stability of mixture BB and mixture F. Gravimetric values were normalised and the peak areas compared. There are two columns for the comparison of mixture BB and CC as the comparison was repeated on two consecutive days."

P14, L427: suggest: "were compared after 1, 8, and 83 days."

Done, sentence now reads: "Following decant of mixture BB into the SilcoNert 2000® treated canister the contents were compared against mixture BB after 1, 8 and 83 days."

P14, L440: This needs more explanation. What was it about the sample flow: too little flow, too little pressure to flush tubing?

Sentence now reads: "This can be attributed to changes in the flow of gas from the canister samples during measurement due to the volume and pressure of gas contained."

P15, L452: An introductory sentence is needed here. This paragraph should begin with the objective: comparing gravimetric PRMs to an SI-traceable dilution method.

The authors agree and have reworked much of Section 3.4 to improve the flow and structure. Opening paragraph now reads: "Two SI traceable preparation techniques for producing reference gas mixtures were compared. One was the preparation of static gravimetric PRMs, the other the generation of a dynamic reference standard from ReGaS2 using a permeation tube. From the weighing of the limonene permeation tube and from the data that was logged for the nitrogen flow and subsequent dilution it was calculated that the ReGaS2 mobile gas generator was outputting $4.41 \pm 0.32$ nmol mol$^{-1}$ of limonene with an expanded uncertainty of 7.3 % ($k = 2$). Using the PRM static  standards gravimetrically produced the output of the ReGaS2 dynamic

system was estimated to be 3.57 ± 0.11 nmol mol⁻¹ of limonene with an expanded uncertainty of 2.9 % (*k* = 2)."

P15, L462: semi-colon after NIST

Semi-colon added.

P15, L466: poor grammar here.

The authors agree and have reworked much of Section 3.4 to improve the flow and structure. Sentence now reads "One of the reasons for the systematic bias between the two approaches can be attributed to the temperature at which the permeator was operated, as the temperature was observed to have a strong influence on the reproducibility of the permeation rate."

P16, L506: This percentage difference should be stated in the previous section. Also, if there were additional complications with the dilution system, the uncertanty assigned is probably too low.

The authors agree; this has been addressed. Paragraph now reads: "The second reason is the 15-20 % decrease in the permeation rate. To investigate this further the permeation rate of limonene from the ReGaS2 dynamic system was measured over an 11 month period between March 2017 and February 2018. The decrease in the permeation rate was determined to be 35 % over this temporal period (Figure S3) for the same temperature. The measurement of the permeation rate in the magnetic suspension balance lasted between two and seven days with an associated uncertanty between 0.5 and 1.5 % for one measurement at one temperature thus suggesting that the uncertainty assigned to ReGaS2 during the comparison was too low."

**Reviewer 2 Comments**

**Main points:**

Lines 210-212: I find this paragraph confusing and after reading it many times I do not think it actually describes what you are doing with the normalised ratio. The crux of it all is the term 'gravimetric difference'. What you really are looking at here is the difference between the amount fraction you would expect in the absence of losses post-decant (in which case it would be equal to the gravimetric amount fraction of cylinder 1 as calculated from the dilution data) and the certified amount fraction obtained from a comparison with mixture BB as you described in lines 194-196. Please re-phrase.

The authors accept some of this terminology is confusing therefore the Section, as well as the Figures and Equations have been altered for clarity. Section reads as follows:

"All of the analyses were performed using GC-FID as described in Section 2.2. The amount fraction of each compound in the decanted cylinder was determined through a comparison with a nominal 2 nmol mol⁻¹ reference PRM (mixture BB). If there were no losses then the amount fraction of the decanted cylinders would be the same as those of the PRM cylinder 1.  Decant losses were determined for each compound by calculating the relative difference between the amount fraction (AF$_{decant}$) of each compound in the decanted mixture and the expected amount fraction of that compound (AF$_{expected}$), which was defined as its gravimetric value before any decanting:

$$\text{relative difference (\%)} = \left( \frac{AF_{decant} - AF_{expected}}{AF_{expected}} \right) \times 100 \qquad (1)$$

The amount fraction of each compound after decanting ($AF_{decant}$) was calculated from:

$$AF_{decant} = \frac{Area_{avgdecant}}{Area_{avgBB}} \times Grav_{BB} \qquad (2)$$

where, $Area_{avgdecant}$ was the average peak area for a set of GC runs (typically five) of the decanted mixture, $Area_{avgBB}$ was the average peak area for a set of GC runs of in-house reference PRM, mixture BB, and $Grav_{BB}$ is the gravimetrically assigned value of the compound in mixture BB."

Line 382 and Figure 5: One of the major outcomes of this study is the stability of the monoterpenes studied in one type of cylinder. This is shown in Figure 5 but this finding is backed up by surprisingly little statistical analysis. "appears […] fairly constant" is the only description accompanying the plots in Figure 5. I feel that the conclusions of this study would be stronger if backed up by a more solid analysis. I would like the authors to add a weighted least-squares fit to the plots in Figure 5 and comment on the magnitude of the gradients in the light of its associated uncertainties from the fit. I would be surprised if this analysis did not support the authors' conclusions, but it needs to be shown. Also crop the x-axis at 80 days.

The authors agree with the Reviewer's comments and have done a weighted least fit squares regressional analysis. Figure 5 has been modified to include this fitting and confidence intervals have been added. The x-axis has also been cropped, as shown below:

[Figure]

The following sentence has been added: "Regression analysis using a least squares fit shows that the gradients for all four monoterpenes are within the measurement uncertainty of zero showing no statistically significant change in amount fraction over the 75 day timeframe."

Figure 5 caption now reads: "The short-term stability of reference PRM (mixture BB) at nominally 2 nmol mol$^{-1}$ compared as a ratio of the area of each monoterpene normalised relative to the n-octane internal standard. Error bars are included to account for the relative standard deviation of the mean ($k$ = 2). The solid lines show the results of a linear least squares fitting routine with the shaded area showing the confidence interval (95 %) of the fit."

Minor points:

Lines 64-68: As you went through the effort of referencing work on terpene sampling, I feel it would be worth mentioning what these techniques actually are. I appreciate that a full account on terpene detection is beyond the scope of this manuscript, but it would be nice to at least mention the techniques used by the papers cited.

The authors are happy to oblige and manuscript now reads: "A variety of techniques have been used for the sampling and analysis of complex mixtures of terpenes including active and passive sorbent tube loading and desorption (Sunesson et al., 1999), canister sampling (Batterman et al., 1998; Pollmann et al., 2005) followed by analysis using gas chromatography mass spectrometry (Birmili et al., 2003; Koch et al., 2000), proton transfer reaction mass spectrometry (Holzinger et al., 2005) or other spectroscopic techniques (Qiu et al., 2017)."

Line 97: this is not the first occurrence of the word 'passivation' in the text. Please move the explanation of its meaning to line 90.

As mentioned by both Reviewers, this was moved.

Line 106: "based on permeation" sounds too vague. Consider changing to something along the lines of "based on the dynamic dilution of limonene from a permeation tube".

Sentence now reads: "The PRM containing  limonene was compared to a new dynamic system based on permeation known as Reactive Gas Standard 2 (ReGaS2) developed by the Federal Institute of Metrology (METAS), (Pascale et al., 2017), that is **based on the dynamic dilution of limonene from a permeation tube** to evaluate any systematic biases between the two different approaches."

Lines 114-115: you need to stress the difference between the 4 PRMs containing monoterpenes and the one containing n-octane. I recommend this edit: "PRMs containing the four monoterpenes, -pinene (both the minus and plus optical isomers), 3-carene, R-limonene and 1,8-cineole, as well as one containing n-octane (used as an internal reference standard), were prepared independently (no comma!) in a balance of high purity…".

Following both Reviewers comments the sentence now reads: "PRMs containing the four monoterpenes, α-pinene (both the minus and plus optical isomers), 3-carene, R-limonene and 1,8-cineole, as well as one containing n-octane (used as an internal reference standard), were prepared independently in a balance of high purity dry nitrogen 115 (BIP+, Air Products) in accordance with ISO 6142 (ISO, 2015)."

Line 116: "high purity dry nitrogen". I feel like the inclusion of "dry" here (as well as in lines 120 and 140) is redundant, especially as the water content is specified further down (line 123).

Word "dry" deleted.

Line 141: No point in repeating the specs of the purifier here (they already appeared in lines 121-122). Just replace with "purifier".

Replaced.

Line 162: connected to what?

Sentence now reads "The PRMs were connected **to the GC** using a minimal dead volume connector and the flow rate was set to 50 mL min$_{-1}$ using a custom flow restrictor."

Line 180 and following: I do not understand the use of the quote marks for the cylinder number here and in the rest of the manuscript. 1, 2 and 3 are the effective labels of the mixtures, just like AA and BB. I would recommend removing these as they are confusing and redundant.

The authors accept that the quotation marks are unnecessary and have been removed from all relevant parts of the manuscript.

Line 202: how many runs were averaged in each set? Give a typical value in the text

This paragraph has been reworked following the Reviewer's earlier remarks. A numerical value has been given "(typically five)".

Lines 324-327: this is very interesting and perhaps deserves a short explanation of why such effects arise at low pressures?

This part has been extended and referenced: Below 30 bar we observe that the ratio is less than 1 for all components. While the results are within the measurement uncertainty, wall factors could have an influence on composition at low pressures (< 30 bar) (Brewer et al., 2018). As reported in Brewer et al. (2018) compounds adsorbed to the walls at high pressure were observed to desorb back into the gas phase at lower pressures."

Lines 362-363: The sentence starting with "A sample of the reference…" is a repetition of the sentence in lines 349-350.

This has been removed. First paragraph has been rephrased to address both Reviewer's suggestions.

Line 396: where does the figure "2.5%" come from? The largest difference in Table 2 is 1.96 %. Perhaps change to "no greater than 2%"?

If the numbers are in modulus terms yes, however if we consider the numbers directionally then it is over 2 %.

Lines 466-473: revise the grammar of this entire paragraph.

The authors agree and have reworked much of Section 3.4 to improve the flow and structure. Paragraph now reads "One of the reasons for the systematic bias between the two approaches can be attributed to the temperature at which the permeator was operated, as the temperature was observed to have a strong influence on the reproducibility of the permeation rate. At lower temperatures, such as 30°C (which was the temperature used for the comparison), the permeator does not reach a true steady state and it was observed that the variability on the permeation rate for the same temperature between two measurements was between 8 and 10 %. A shift in the permeation rate of this magnitude coupled to uncertainties in temperature would be enough to compensate for the systematic bias observed between the two approaches."

Table 2: explain in the table caption why there are two columns for BB v CC.

Table 2 caption has been clarified to "Comparison showing the percentage difference between PRM mixtures prepared 389 more than two years apart to assess the long-term stability of mixture BB and mixture F. Gravimetric values were normalised and the peak areas compared. There are two columns for the comparison of mixture BB and CC as the comparison was repeated on two consecutive days."

Figures 2, 3, 4, 5: These all seem to be at a lower resolution than Figures 1 and 6. Would the authors provide higher resolution figures? The better resolution helps a lot, especially when zooming in on the electronic version of the paper.

In the final submission of the manuscript the resolution of all of the Figures should be higher as they will be submitted as .eps files that can save vector information and improve the resolution.

Figure 4: the caption needs to actually introduce what the figure represents. Start off with "Typical chromatogram of…"

Opening line of Figure caption now reads: "Typical chromatograms for a stable (pink) and an unstable (grey) terpene mixture."

Tables S2-S7: in the captions, state what the percentage difference refers to. The response factors? The normalised ratios?

Captions now specify "The percentage difference between the reference standard mixture BB and the decanted monoterpene mixture in a 10 L internally passivated XX cylinder (normalised for gravimetric differences)."

Tables S2 and S3: data in S3 is the repeat of S2, but the cylinder types are labelled differently. I understand that Experis and Quantum refer to the same thing, but it would make things easier to just stick to one name (and it would also be consistent with the following tables).

Well spotted, this has been rectified.

Tables S4-S7: in the captions, state why there are missing entries in these tables.

Each table now has the phrase "No further decants were performed for this cylinder type as the passivation was shown to be unsuitable for monoterpenes." Placed inside the table where the missing entries are.

Typographical, grammar and language corrections:

The citation style is not in the AMT recommended format. Revise throughout.

Apologies, the library was formatted in the wrong style – this has been corrected to be consistent with AMT style.

Shouldn't all equations be numbered?

They are now.

Line 43: avoid double parentheses when possible e.g., a hemiterpene of formula C5H8

This had been left as the authors do not think that it is necessary to change.

Lines 78-79: This sentence sounds like a fragment of a larger sentence. I suspect it should be part of the previous sentence. Please rephrase. Also, change 'prevent the reference material dominating' to 'prevent the reference material from dominating'.

Sentence modified to "In order to meet the 5 % uncertainty target and **prevent the reference material from dominating** the uncertainty requires stable PRMs of monoterpenes with uncertainties of better than 1.25 % (less than a quarter of the uncertainty)."

Line 163: replace 'ml' with 'mL' (occurs twice in this line)

Changed.

Line 175: no need for the comma after Quantum.

Comma removed.

Line 194: Replace 'was' with 'were'

Corrected.

Line 195: Replace 'valve' with 'value'

Changed.

Line 204: add a comma after BB.

Done.

Lines 207-208: remove the comma after "where" and add a comma after BB.

Removed.

Line 228: replace 'run' with 'analysed' or similar.

Sentence now reads: "Over the first three month period that mixture BB was **analysed** the ratio of the monoterpene to *n*-octane remains constant within the measurement uncertainty."

Line 234: response factors (plural)

Changed.

Line 256: replace with "can produce traceable reference gas mixtures of a number of species, including terpenes"

Sentence now reads: "The ReGaS2 is a mobile generator that **can produce traceable reference gas mixtures of a number of species, including terpenes** (Pascale et al., 2017)."

Lines 258-263: the verb tenses are all mixed up (past, present). It makes following the text harder than it should be. Please pick one tense and stick to it.

Paragraph now reads: "The method is based on permeation and subsequent dynamic dilution: a permeation tube containing the pure terpene is stored in an oven used as permeation chamber. The pure substance permeates at a constant rate into the matrix gas and can be diluted to give the desired amount fraction. The mass loss over time of the permeation tube is precisely calibrated using a traceable magnetic suspension balance. All parts in contact with the reference gas are coated with SilcoNert2000®."

Lines 267-268: remove "as was"

Removed.

Lines 353: replace semicolon with colon

Replaced.

Line 488: change to "systems in the field".

Done.

Line 496: I am not a huge fan of the word 'dependency'. 'Dependence' is more commonly used when describing the relationship of a variable to another (temperaturedependence, pressure-dependence, and so on).

Word deleted. Sentence now reads "We have demonstrated that Experis treated cylinders are the most appropriate for containing low amount fraction monoterpene PRMs and that the amount fraction is not influenced by pressure between 30 and 120 bar."

Tables S2-S7: In one of the headers, change "difference to" with "difference with respect to"

Changed.

Figure S1: in the caption, change spectrum to spectra and remove the comma after terpinolene.

Done, caption now reads: "Mass spectrometry ion fragmentation spectra for (a) camphene (b) $\alpha$-terpinene (c) cymene (d) $\tau$-terpinene (e) terpinolene peaks identified and observed in a BOC SPECTRA-SEAL passivated cylinder."

---

## Author Response (AR2)

Dear Authors,

Very nice job in addressing the Reviewers' suggestions. There is one minor thing (sentence) that I believe should be changed - contained in lines 239 - 241.

"It has been well documented that the use of stainless steel canisters for sampling terpenes in dry or humidified air can be problematic (Batterman et al., 1998; Rhoderick, 2010)."

First of all the Rhoderick study did not address the use of SS canisters - all it did was reference the Batterman study within. Also, the Batterman study used one type of SS canister. It is possible that other e-polished canisters could do a better job. We don't know but my point is that saying "it is well documented" is a stretch. I suggest revising the sentence and dropping the Rhoderick reference.

Best,

Eric Apel

Dear Eric.

Thank you for your kind response and the thoroughness of your review. We have removed the Rhoderick reference and modified the sentence to "Previous work has shown that the use of stainless steel canisters for sampling terpenes in dry or humidified air can be problematic (Batterman et al., 1998)." I hope that this suffices.

Thanks for your help.

Nick (Allen)